# The Influence of Selected Factors on the Nutritional Value of the Milk of Cold-Blooded Mares: The Example of the Sokólski Breed

**DOI:** 10.3390/ani13071152

**Published:** 2023-03-24

**Authors:** Joanna Barłowska, Grażyna Polak, Iwona Janczarek, Ewelina Tkaczyk

**Affiliations:** 1Department Quality Assessment and Processing of Animal Products, University of Life Sciences in Lublin, Akademicka 13, 20-950 Lublin, Poland; 2Office of the Director for Scientific Affairs, National Research Institute of Animal Production, Krakowska 1, 32-083 Balice, Poland; 3Department of Horse Breeding and Use, University of Life Sciences in Lublin, Akademicka 13, 20-950 Lublin, Poland

**Keywords:** mare’s milk, nutritional value, pasture, lactation number, foal gender

## Abstract

**Simple Summary:**

Mare milk, as well as fermented products produced from it, are recognized as having medicinal effects. Mare milk has been consumed for hundreds of years in Central Asia and Eastern Europe. In Europe it became popular in the 1990s. However, in many European countries, including Poland, the milking of mares remains marginal. Nevertheless, the group of consumers interested in purchasing mare milk is gradually growing. The horse population in Poland is about 300,000, of which half are cold-blooded horses. Native breeds of cold-blooded horses, including the Sokólski horse, are included in a genetic resources conservation programme. These horses are used in diverse ways, including for meat. For ethical and cultural reasons, however, the slaughter of horses is negatively perceived in Poland. An alternative to this type of use could be the acquisition of milk from mares. Therefore, the aim of the study was to determine the effect of selected factors (access to pasture, lactation number, and sex of the foal) on the proximate composition, whey protein profile, and fatty acid profile of milk from mares of the Sokólski breed. The results may be useful for those managing herds of Sokólski mares and other mares of cold-blooded breeds that are in good condition.

**Abstract:**

This study assessed the effect of access to pasture, lactation number, and foals’ sex on the nutritional value of milk (79 samples) from nine mares. The following were analysed: content of dry matter, protein, fat, lactose, and ash; percentages of α-lactalbumin (α-La), β-lactoglobulin (β-Lg), serum albumin (SA), immunoglobulins (Ig), lactoferrin (Lf), and lysozyme (Lz) in the total protein; and the fatty acid profile. Mares without access to pastures were shown to produce milk with a higher dry matter content, including fat, lactose, and ash; higher percentages of β-Lg, α-La, Ig, and Lf; and a better fatty acid profile. The milk from mares with access to pasture contained more protein, including higher percentages of SA and Lz. Milk from mares in lactations 4–6 had the highest fat and protein concentrations and the lowest lactose concentration. The α-La level was highest in lactation 1, Lf in lactations 2–3, and Lz in lactations 4–6. Milk from mares in lactations 4–6 had the best fatty acid profile (the lowest concentration of saturated fatty acids (SFAs) and the highest concentration of monounsaturated fatty acids(MUFA) and polyunsaturated fatty acids (PUFA)). Milk from mothers of female offspring had higher dry matter, fat, and protein concentrations, a higher share of lysozyme, and a better fatty acid profile.

## 1. Introduction

According to the Food and Agriculture Organization (FAO) [1], there are 905 registered horse breeds, of which 124 are transboundary breeds (63 international transboundary) and 694 are local breeds. Among all these breeds, the risk status of 104 is critical, while 10 are critical-maintained, 67 are endangered, 21 are endangered-maintained, 157 are not at risk, and 87 are extinct. In Poland, breeding work is conducted on 10 horse breeds [2], of which seven are included in a genetic resource conservation programme (warm-blooded Malopolski, Wielkopolski and Silesian; primitive Hucul and Polish Konik; and cold-blooded Sztumski and Sokólski) [3]. The Sokólski horse was the latest to come under protection, in 2008 [4]. Apart from the traits qualifying the breed for conservation, it is worth noting its versatility, including its potential use for milking [5].

Published research results show that it is mainly cold-blooded mares that are predisposed to milking, due to their even temperament, gentle nature, and willingness to work with people [6]. In addition, milk production from these breeds is usually higher than in warm-blooded horses [7]. These factors suggest that the Sokólski horse could be used for this purpose to a greater extent than at present.

The Sokólski horse type was created in the 19^th^ century in what was then Northeast Poland, from local mares improved with imported Breton and Ardennais stallions. In 1928 two Artillery Horse Breeders Clubs were established—one in Sokółka and the other in Janów. After World War II, breeding work at the Sokółka centre led to the creation of one of the most genetically and phenotypically consolidated types of cold-blooded horses in Poland. However, only conservation of the genetic resources of this breed has allowed its population to gradually increase from its drastically low level at the start of the 21st century [3,5,6].

The temporary range of uses for horses is very wide, including work as draught horses, riding for sport or recreation, and use for meat or milk production [8]. There is a centuries-long tradition of milking mares in Central Asia and Eastern Europe. Milk from these animals is one of the basic component parts of everyday diet items for human populations living in these regions. However, it can sometimes cause diarrhea, so it is not consumed immediately after milking [9]. It is generally subjected both to lactic acid and alcoholic fermentation, resulting in the famous koumiss. Both mare milk and koumiss are recognized as medicinal products in Bashkortostan, Kazakhstan, Uzbekistan, and Ukraine [10]. In contrast, in most countries of contemporary Europe horse milk was until recently forgotten, although up to the late 1950s it was used as an aid in the treatment of gastrointestinal diseases (gastric ulcers, cirrhosis, cholecystitis, and pancreatitis), respiratory diseases (tuberculosis, bronchitis, pertussis, and asthma), and migraines [11].

There is currently a renaissance of interest in horse milk and milk products [8,12]. The production and processing of mare milk in Europe is developing mainly in France, Italy, Greece, and Germany. It is valued in China as well [13]. According to Musaev et al. [14], mare milk has antibacterial and antiviral properties. It can be used in the treatment of tuberculosis, hepatitis C, and psoriasis. In Russia, fermented mare milk (koumiss) is used in the treatment of gastrointestinal and cardiovascular disease [15]. An estimated 30 million people around the world regularly consume mare milk [16]. Nevertheless, in many European countries, including Poland, the milking of horses for drinking milk remains a marginal practice [11]. However, a growing group of consumers is interested in purchasing it.

Mare milk differs significantly in composition from cow milk and is similar to human milk [16,17]. It contains about a third less fat than the milk of ruminants and humans [16]. In addition, there are significant differences in the fatty acid profiles of horse and cow milk, due to the structure of the gastrointestinal tract in these species [18]. Mare milk also has a lower protein content than cow milk. The content of casein in mare milk (1.07%) is less than half that of cow milk (2.51%), but is three times higher than in human milk (0.37%) [9]. The percentage of whey proteins is much higher in mare milk (about 40%) and human milk (about 50%) than in cow milk (about 20%). For this reason, mare milk and human milk are referred to as albumin milk, while cow milk is called casein milk [10]. It is also worth noting that mare milk (like human milk) has a high content of lactose (about 7%), compared to about 4.8% in cow milk [17].

The composition and nutritional value of the milk of dairy animals depends on genetic factors (breed and individual traits), physiological factors (stage of lactation, age) and environmental factors (diet, time of year) [19,20,21,22,23,24,25,26]. In the case of horses as well, research indicates that the composition of milk is influenced by the breed, stage of lactation, parity, and diet [18,27,28,29,30,31]. We assumed that horses’ sensitivity to various factors may be modified by breeding work on the species and changes in the maintenance of the animals. This may mean that studies on the effect of various factors on the use value of horses should not only be repeated, but should be analysed for specific populations. It also seems important to include new factors in the analysis. As equids are polygynous animals [32], one such factor may be the sex of the foal reared by the mare.

Therefore, the aim of the study was to assess the effect of selected factors (access to pasture, lactation number, and sex of the foal) on the proximate composition, whey protein profile, and fatty acid profile of milk from mares of the native Sokólski breed.

## 2. Materials and Methods

### 2.1. Animals and Research Material

#### Horses

The study was conducted using 9 mares of the cold-blooded Sokólski breed, kept in a herd at the Experimental Station of the National Research Institute of Animal Production in Kołbacz (IZ PIB). The mares were milked regularly. Each of the mares had given birth to its first foal at the age of 4 years. The study was conducted in the years 2017–2020. The mares were in lactations 1, 2, and 4 in 2017, 1–4 in 2018, 1, 2, 4, and 5 in 2019, and 1, 2, 5, and 6 in 2020. In total, 7 colts and 8 fillies were reared by the mares (2017—2 colts and 2 fillies, 2018—2 colts, 2019—2 colts and 2 fillies, 2020—1 colt and 4 fillies). The mares were kept in a stable with their foals in individual 4 × 4 m boxes. Each stall was equipped with a corner trough for the mare, a trough for the foal, a NaCl salt lick, and an automatic drinker. In 2017–2018 they had access only to a paddock without grass, but in 2019–2020 they were able to use a pasture seasonally. The botanical composition of the pasture included about 20% tall grasses (mainly perennial ryegrass and meadow fescue), 50% short grasses (mainly Kentucky bluegrass), 15% white clover, and numerous herbs. During the period when they were milked, the mares were let out of the stable between 11:00 to 12:00 and brought back at 18:00. Outside of this period, the mares were let out of the stable at about 7:30. The mares were fed twice a day, at 6:00 and 18:30. They had unlimited access to water 24 h a day. The composition of the diets is presented Table 1.

### 2.2. Research Material

The research material was milk obtained individually from mares beginning from the 60th day after foaling. The mares were milked once a day between 11:00 and 12:00. Before milking, the foals were separated from their mothers for about 3–4 h and led to stalls adjacent to the milking stalls to enable visual and tactile contact with their mothers. Based on the experience of one of the authors (GP), as well as the gentle nature and even temperament of the mares and foals, the absence of a negative reaction to the milking process and temporary separation, and the contact ensured by the adjacent stalls, it was decided that there was no need to accustom the animals to milking. In addition, to compensate for any stress response, the mares received an additional portion of crushed oats and carrots during milking (Table 1), while the foals received crushed oats ad libitum. The average milking period for the mares was 113 days. Each milking session was preceded by forestripping, an udder massage, and teat disinfection. The milk was drawn in full from both halves of the udder using an Alfa-Laval milking machine adapted for horses. Milk samples were collected from each mare 3–4 times at 3–4-week intervals during lactation, whenever possible (in some cases the mares withheld their milk). In total 79 milk samples were collected (in 2017—29, 2018—19, 2019—10, and 2020—21 samples). The samples were collected into 400 mL bottles and then cooled and transported to the laboratory.

### 2.3. Analytical Methods

In the laboratory, each sample was divided into two portions: the first portion (for basic composition) was not frozen, and the remainder (for analysis of the whey protein and fatty acid profile) was frozen in two 40 mL containers. The milk samples were defrosted prior to laboratory analyses. Each milk sample was analysed for its basic chemical composition (content of dry matter, protein, fat, lactose, and ash); whey protein profile, i.e., the percentages of α-lactalbumin (α-La), β-lactoglobulin (β-Lg), serum albumin (SA), immunoglobulins (Ig), lactoferrin (Lf), and lysozyme (Lz) in the total protein; and fatty acid profile (concentrations of individual acids). The analyses were performed in duplicate for each milk sample. The crude protein content was determined by Kjeldahl’s method in a Buchi apparatus according to PN-EN ISO 8968-3:2008 [33]; fat content by Gerber’s method according to PN-ISO 2446:2010 [34]; dry matter by the oven-dry method according to PN-ISO 6731:2014-11 [35]; ash by mineralization according to Association of Official Agricultural Chemists (AOAC) 1995 [36]; and lactose according to AOAC 935.42:1995 (Bertrand method) [37].

The whey protein profile was determined by separating the protein fractions by gel electrophoresis (SDS-PAGE) ). To this end, 0.5 mL of milk was dissolved in 1 mL of reducing buffer (0.5 M Tris-HCl with pH 6.8, glycerol, 10% SDS, 0.5% bromophenol blue, β-mercaptoethanol, and deionized water) and heated at 95 °C for 4 min. A 5 μL volume of sample prepared in this manner was applied to polyacrylamide gel. Protein standards in the same volume were placed in the edge wells. Separation was carried out at 100 mV on a 12% SDS-PAGE polyacrylamide gel using the BIORAD Mini-PROTEAN 3 Cell system according to Laemmli [38]. The gels were stained in R 250 Coomassie solution and then scanned and subjected to quantitative analysis using Gelscan v. 2.0 software for analysis of the electrophoretic gels (Krzysztof Kucharczyk Techniki Elektroforetyczne Sp. z o.o., Warsaw, Poland). The content of whey proteins in the milk was confirmed by liquid chromatography following precipitation of casein at pH 4.2 [39] using the Dionex Ultimate 3000 chromatograph (Thermo Fisher Scientific, Waltham, MA, USA), equipped with an autosampler, a PLRP-S column (4.6 mm i.d. × 150 mm, 5 μm, 300 Å from Polymer Laboratories, Shropshire, UK), and a 205 nm UV VIS detector. Chromeleon v. 7.0 software (Thermo Fisher Scientific, Waltham, MA, USA) was used for the calculations.

The fatty acid profile was determined following the extraction of milk fat according to AOCS Official Method Ce 2-66 [40]. To this end, about 15 mL of milk was collected into a plastic container (Sarstedt, Nümbrecht, Germany). Chloroform and methanol were added in a 2:1 volume ratio in the amount of 30 mL. Then the sample was homogenized for 5 min. After a 5-min interval, the sample was homogenized again for 5 min. Next, the sample was centrifuged at room temperature (7000 rpm for 5 min). The chloroform layer was removed, filtered through anhydrous sodium sulphate, and dried in a nitrogen stream at 45 °C. The extracted milk fat was esterified. For this purpose, 5 mg of fat was dissolved in 50 µL of toluene, and 100 µL 2 M of methanolic sodium hydroxide solution was added. Esterification was carried out at room temperature for 20 min. Then, 0.5 cm^3^ of 14% boron trifluoride-methanol solution was added, and the sample was left at room temperature for 20 min. Fatty acid methyl esters (FAME) were extracted twice using 2 cm^3^ of hexane. The samples prepared in this manner were subjected to chromatographic analysis according to PN-EN ISO 12966-1:2015-01 [41] and PN-EN ISO 5508 [42] using a TRACE 1300 chromatograph (Thermo Fisher Scientific, Waltham, MA, USA) with a flame ionization detector (FID) equipped with a BPX-70 column (60 m × 0.25 mm × 0.20 mm). The carrier gas was helium (5 mL/min), and the split flow rate was 10 mL/min. The FID temperature was 250 °C, and the injector temperature was 220 °C. The temperature programme was: from 60 °C (held for 3 min) to 200 °C, with increments of 7 °C/min, held at 200 °C for 20 min. Fatty acids were identified by comparing the retention times of fatty acid methyl esters (FAME) with a mixture of Supelco 37 FAME Mix standards and PUFA No. 2 (Animal Source) (Sigma-Aldrich Co., St Louis, MO, USA). The fatty acid profile was determined by identifying and calculating the relative peak areas. The results were expressed as percentages of the individual fatty acids in the total amount of methyl esters.

On this basis, the following groups of fatty acids and their proportions and indices were calculated:SFA—sum of saturated fatty acids.MUFA—sum of monounsaturated fatty acids.PUFA—sum of polyunsaturated fatty acids.n-6 and n-3.n-6/n-3.DFA—desirable fatty acids (MUFA + PUFA + C18: 0) according to Medeiros et al. [43].HSFA—hypercholesterolaemic saturated fatty acids—$\left(C12:0+C14:0+C16:0\right)$—according to Renna et al. [44].AI—(atherogenicity index) = $\left(C12:0+4\times C14:0+C16:0\right)÷\left(MUFA+PUFA\right)$ according to Ulbricht and Southgate (1991) (qtd. in [45]).TI—(thrombogenicity index) = $\left(C14:0+C16:0+C18:0\right)÷\left(0.5\times MUFA+0.5\times n6+3\times n3+n3÷n6\right)$ according to Ulbricht and Southgate (1991) (qtd. in [44]).

### 2.4. Statistical Analysis

The distribution of traits was analysed using the Kolmogorov–Smirnov, Cramér–von Mises, and Anderson–Darling tests at α = 0.05.

The significance of the effect of the fixed factors on the analysed traits was verified by multivariate analysis of variance (General Linear Models (GLM) procedure) using the following model:Y_ijkmn_ = μ + w_i_ + p_j_ + l_k_ + z_m_ + e_ijkmn_,
where: Y is a given milk trait, μ is the average for the trait, w is the fixed effect of pasture (i = 2: pasture (mares with pasture access), no pasture (mares without access to pasture)), p is the fixed effect of the sex of the offspring (j = 2: filly, colt), l is the fixed effect of the lactation number (k = 3: lactation 1, lactation 2–3, lactation 4–6), z is the fixed effect of the year of research (m = 4: 2017, 2018, 2019, 2020), and e is the residual error.

The significances of the differences between means were determined using Tukey’s multiple comparisons (for non-orthogonal data) with Bonferroni’s correction and presented in tables with the standard deviation (SD) and standard error of the mean (SE). Means designated with different capital letters (A, B) are significantly different at α = 0.01; lower case letters (a, b) indicate significance at α = 0.05.

The statistical calculations were performed using the SAS 9.4 analytical software package (version 9.4 by SAS Institute Inc. Cary, NC, USA).

## 3. Results

The analysis of the effect of the type of bulk feed used in the diet (with vs without pasture) showed that milk from mares without access to the pasture had a significantly (*p* ≤ 0.05) higher content of dry matter (by 0.10%), including fat by 0.12% (*p* ≤ 0.01), lactose by 0.13%, and ash by 0.01% (*p* ≤ 0.05), in comparison to mares with access to pasture. On the other hand, the protein concentration of milk from mares using the pasture was 0.16% higher (*p* ≤ 0.01; Table 2). The milk from mares without pasture access had a higher percentage of whey proteins in the total protein (by 2.96%), including β-lactoglobulin (by 2.97%), α-lactalbumin (by 0.71%), immunoglobulin (by 0.81), and lactoferrin (by 0.34%), while milk from mares with pasture access had higher levels of serum albumin (by 0.44%) and lysozyme (by 0.8%; Table 3). The analysis of the fatty acid profile showed that milk from mares with pasture access had significantly (*p* ≤ 0.01) higher levels of saturated fatty acids (SFA) and lower levels of monounsaturated fatty acids (MUFA), with similar levels of polyunsaturated fatty acids (PUFA; Table 4). The higher share of SFA in this milk was due mainly to the content of stearic acid (C18:0), which was more than 3.5 times as high as in the milk from mares without pasture access. In addition, the milk from mares with access to pasture had higher (*p* ≤ 0.01) concentrations of acids C4:0, C6:0, C8:0, C17:0, and C20:0 (Table 4). The higher MUFA content in milk from mares without pasture access was determined by the levels of acids C14:1, C16:1 n-9, C16:1 n-7, and C18:1 n-9. Milk from these mares also had a higher (*p* ≤ 0.01) concentration of linoleic acid LA (PUFA C18:2 n-6). Use of the pasture, on the other hand, increased the share of γ-linolenic acid GLA (PUFA C18:3 n-6), α-linolenic acid ALA (PUFA C18:3 n-3), and conjugated linoleic acid CLA (PUFA C18:2 cis9 trans11). The level of n-6 PUFAs was significantly (*p* ≤ 0.01) higher in milk from mares without pasture access, while that of n-3 PUFAs was significantly higher in milk from mares with pasture access (Table 4). The ratio of these acids (n-6/n-3), however, was lower (more favourable) in the milk from mares with pasture access (Table 5). No significant differences depending on this factor were noted for the remaining parameters characterizing the fatty acid profile of milk (Table 5).

The analysis of the effect of lactation number on the proximate composition of the milk showed that the protein content successively increased with each lactation (*p* ≤ 0.05); the difference between lactation 1 and lactations 4–6 was 0.18% (Table 2). The highest fat content was also noted during lactations 4–6, and the lowest in lactations 2–3 (*p* ≤ 0.05); the difference was nearly twofold. The reverse pattern was observed for lactose content (*p* ≤ 0.01), which was highest during lactations 2–3 and lowest in lactations 4–6 (Table 2). The highest percentage of whey proteins in the total protein was noted in the first lactation, although this was not confirmed statistically (Table 3). This was due to the fact that the milk from this lactation had the highest percentages of the two major proteins in the whey fraction, β-lactoglobulin and α-lactalbumin, although this was confirmed statistically only for the latter (*p* ≤ 0.05). In the case of proteins with bactericidal properties, the lactoferrin content was highest in the milk from lactations 2–3 and lowest for lactations 4–6 (*p* ≤ 0.01). The reserve pattern was observed for the content of lysozyme, which was highest during lactations 4–6 and lowest during lactations 2–3 (*p* ≤ 0.01). No significant differences were shown between successive lactations for the share of immunoglobulin and serum albumin (Table 3). The analysis of the fatty acid profile of the milk showed that multiparous mares (lactations 4–6) produced milk with the lowest (*p* ≤ 0.01) level of SFAs and the highest level of MUFAs (*p* ≤ 0.05) and PUFAs (*p* ≤ 0.01), including n-3 acids (*p* ≤ 0.01; Table 4). The lower level of SFAs in this group of mares (lactations 4–6) was mainly due to the significant (*p* ≤ 0.01) decrease in the content of fatty acids C10:0, C14:0, and C16:0. The increased content of MUFAs in the milk of multiparous mares was mainly due to the significant (*p* ≤ 0.01) increase in the concentration of oleic acid (C18:1n-9), and in the case of PUFAs, a significant (*p* ≤ 0.01) increase in the content of α-linolenic acid ALA (PUFA C18:3n-3; Table 4). Milk from mares in lactations 4–6 had the highest (*p* ≤ 0.01) share of DFAs and the lowest (*p* ≤ 0.01) share of HSFAs, as well as the lowest (*p* ≤ 0.01) values for both the atherogenicity index (AI) and thrombogenicity index (TI) (Table 5).

The milk from mares with female offspring had a significantly (*p* ≤ 0.01) higher content of dry matter, by 0.11%, including protein, by 0.21% (*p* ≤ 0.05), and fat, by 0.13% (*p* ≤ 0.01), but a lower content of lactose, by 0.23% (*p* ≤ 0.01; Table 2). In the case of whey proteins, the milk from mares with male offspring had a higher (*p* ≤ 0.01) content of lactoferrin (by 0.46%) and serum albumin (by 0.27%), while the milk from mares with daughters had a higher (*p* ≤ 0.01) share of lysozyme (by 1.47%; Table 3). Milk from mares with female offspring had a significantly (*p* ≤ 0.01) lower concentration of SFAs (by 3.708 g/100 g of fat) and a higher (*p* ≤ 0.01) content of PUFAs (by 2.681 g/100 g of fat), including (*p* ≤ 0.01) n-3 acids (by 1.949 g/100 g of fat; Table 4). Mares with female offspring produced milk with a lower (more favourable) n-6/n-3 ratio (*p* ≤ 0.01) and a higher concentration of desirable fatty acids (DFA), by 4.125 g/100 g of fat (*p* ≤ 0.01), with a lower content of hypercholesterolaemic saturated fatty acids (HSFA), by 4.668 g/100 g of fat (*p* ≤ 0.01). In addition, this milk had significantly (*p* ≤ 0.01) lower values for the atherogenicity index (AI), by 0.263 g/100 g of fat, and the thrombogenicity index (TI), by 0.210 g/100 g of fat (Table 5).

## 4. Discussion

The results of the study indicate that each of the factors analysed (access/lack of access to pasture, lactation number, and sex of the foal) influenced the proximate composition, whey protein profile, and fatty acid profile of the mares’ milk.

The results pertaining to the chemical composition of milk from mares with and without access to pasture are difficult to compare with the findings of other authors, as we were unable to find published studies with horses as the experimental animal. This is most likely because the use of horses for milk on a global scale remains marginal. In the case of ruminants, the physiology of digestion and the effect of feedstuffs and additives used in the diet on the composition and nutritional value of milk are well known. In the case of cattle (the most important species in world for milk production), research results indicate that the use of complete feeds (without pasture) improves the yield and proximate composition of milk (higher concentrations of dry matter, fat, and protein, including casein) in comparison to cattle with access to pasture [19,23,25]. The use of a traditional diet with pasture forage, on the other hand, has a beneficial effect on the fatty acid profile (higher share of unsaturated fatty acids, both MUFAs and PUFAs) [19,26,46] and increases the content of whey proteins, i.e., β-lactoglobulin, α-lactalbumin, lactoferrin, and lysozyme, and fat-soluble vitamins A, E, and D3 [23,26]. However, these results cannot be directly applied to horses due to differences in the structure of the digestive tract and the physiology of digestion, which determine the quantity and quality of synthesized milk components [18].

A shared trait of horses and ruminants is the fact that they are grassland herbivores adapted for the consumption of large amounts of fibre-rich feed [47,48]. Both groups of animals have fermentation chambers which are colonized by populations of microbes, but in even-toed ungulates, the fermentation processes take place in the anterior part of the gut (the rumen), prior to digestion and absorption in the intestine, thereby increasing the utilization of fermentation products [49]. In odd-toed ungulates, it is the caecum and colon that function as a fermentation chamber [47,49]. The physiology of the equine digestive system is therefore distinguished by a rapid gastric passage (a relatively small stomach), rapid but intensive enzymatic transformation in the small intestine, and long and intensive microbial fermentation in the large intestine [48]. According to Santos et al. [48], the caecum environment in horses cannot be compared to the rumen in ruminants, because the substrate reaching the caecum largely depends on the pre-caecal digestibility of feedstuffs. This means that cytoplasmic proteins (nitrogen) and soluble sugars reach the large intestine in small quantities. However, most carbohydrates from the cell wall and the nitrogen associated with them reach the large intestine, due to the very low hydrolysis of these nutrients in the stomach or pre-caecal environment.

The milk from mares without access to pasture was also shown to have a higher percentage of dry matter, including fat. In the case of cattle, Barłowska et al. [50] showed that Simmental cows produced milk with a higher percentage of dry matter (by 0.6%), including protein (by 0.43%) and fat (by 0.4%), during the period when they were fed preserved bulky feed (haylage + maize silage) compared to the pasture season. Similar relationships were observed by Brodziak et al. [20] in milk from Polish Red and White-Backed cows. Milewski et al. [51] reported the same tendency for goat milk. Mazhitova and Kulmyrzaev [52], in an analysis of milk from 25 Kyrgyz native breed mares, whose diet was based on pasture without any form of concentrate supplementation from May to August, showed that the content of dry matter in the milk during this period ranged from 10.93% to 11.76%, protein 2.20–2.39%, fat 1.29–1.90%, and ash 0.30–0.46%. The access or lack of access to pastures also significantly (*p* ≤ 0.01) influenced the whey protein profile of mares’ milk. In contrast to the results of research on cattle, mares that did not use the pasture produced milk with a higher proportion of whey proteins in the total protein, including a higher content of total albumins, β-lactoglobulin, α-lactalbumin, immunoglobulin, and lactoferrin. Brodziak et al. [20] performed similar analyses in Polish Red and White-Backed cattle and showed that during the period when the cows used the pasture, they produced milk with a higher proportion of whey proteins, including β-lactoglobulin, α-lactalbumin, immunoglobulin, and lactoferrin. Similar findings have been reported by other authors [19,21,24,26,53]. The results of the present study pertaining to the influence of pasture forage on the fatty acid profile are supported by Tanhuanpää and Knudsen [54]. These authors analysed the milk of 29 mares of the American trotter race, of which nine were fed in the stable on hay and a cereal mixture, and 20 grazed on a pasture, and showed that the milk of the latter group had a higher content of the acids C10:1 and C18:3 and a lower content of the acids C18:1 and C18:2, which was confirmed in the present study. Rutkowska et al. [18] observed similar patterns in a comparison of the fatty acid profile of milk from cold-blooded mares obtained in winter, when the diet consisted of hay, oats, and oat straw, and in summer, when the diet contained a large share of grass and a small amount of oats. Mares using the pasture produced milk with a higher level of short-chain saturated fatty acids (SCSFA), a smaller amount of monounsaturated fatty acids (MUFA), and a similar share of polyunsaturated fatty acids (PUFA). The higher level of SCSFAs during the pasture period was determined mainly by caprylic (C8:0) and decanoic (C10:0) acids, while the smaller MUFA level was determined by palmitoleic (C16:1) and oleic (C18:1 9c) acids. Among PUFAs during this period (summer), the concentrations of linoleic acid (C18:2 9c12c) and CLA (C18:2 9c11t) were slightly higher, while that of α-linolenic acid (C18:3 9c12c15c) was lower.

The second factor analysed, lactation number, was shown to significantly influence the proximate composition of the mares’ milk, the whey protein profile, and the fatty acid profile, which is supported by the findings of other authors [28,55]. Barreto et al. [55] evaluated milk from Quarter Horse mares in successive lactations (1–6) and found that the fat content ranged from 0.61% to 0.84%, with the lowest values noted during the third lactation and the highest during the fourth. Similar patterns were observed in the present study. On the other hand, Czyżak-Runowska et al. [29] reported that mares up to the seventh lactation produced milk with a higher fat content (1.21%), while in subsequent lactations its content fell to 1.06%. In the present study, the protein content in the milk increased in successive lactations. Barreto et al. [55], however, found that the protein content increased gradually up to the third lactation (from 1.62% to 1.74%) but then fell after the fourth foaling (1.61%) and increased again in subsequent lactations (1.74% in lactation 6). The results of the present study are therefore partially in agreement with those cited. Czyżak-Runowska et al. [29] observed a highly stable fat content in successive lactations. Like Barreto et al. [55], we observed that the lactose content was highest in the milk produced by mares in lactations 2–3 and decreased significantly in the milk of older mares. Czyżak-Runowska et al. [29] also reported a significant decrease in the lactose content in the milk of older mares (after the seventh foaling). In the present study, there was a significant decrease in the concentration of α-lactalbumin and an increase in the antimicrobial proteins—lactoferrin and lysozyme—in the milk of older mares. To the best of our knowledge, there have been few studies of this type on horses. Pecka et al. [56] assessed the colostrum and milk (3 and 6 weeks after foaling) of 12 Arabian mares and showed that the content of serum albumin, α-lactalbumin, and β-casein increased slightly (but not significantly) compared to their levels in the colostrum and remained similar at 3 and 6 weeks. The level of class G immunoglobulins, however, decreased significantly (*p* ≤ 0.05) at 3 weeks after foaling relative to the level in the colostrum and then increased slightly (non-significantly) at 6 weeks. In cow milk, the content of β-lactoglobulin and α-lactalbumin has been shown to decrease in successive lactations [22]. This is accompanied by an increase in the content of serum albumin and the two main antimicrobial proteins, lactoferrin and lysozyme, which is linked to an increased somatic cell count. According to the authors, serum albumin is an indicator of the permeability of the blood–milk barrier in the mammary gland. They suggest that this may be due to prior inflammation and the age of the cows, which affects the permeability of epithelial cells of the mammary alveoli. The most favourable fatty acid profile noted in the present study in the oldest mares (lactations 4–6) is confirmed by Czyżak-Runowska et al. [28]. The authors showed statistically significantly higher MUFA concentrations in the milk from mares which had foaled more than seven times than in the milk from mares with lower parity. However, they found no relationship between the parity and the PUFA level. In addition, they found that the atherogenicity index (AI) was significantly lower (more favourable) in older mares (>7 births), which was also observed in the present study.

Female mammals incur high energy costs of reproduction, mainly for lactation [57]. Milk synthesis requires mobilization of the body’s reserves to feed the developing young. The present study showed that mothers of fillies produce milk with a more favourable proximate composition and fatty acid profile, with a similar whey protein profile to that of milk from mothers of colts. Therefore, the results of the present study are similar to those obtained by Hinde et al. [57] in a study using cattle. The authors analysed 2,390,000 lactation records for 1,490,000 dairy cows of the Holstein breed, which has undergone many years of selection for milk production. They showed that cows favour daughters, producing significantly more milk for them than for sons throughout lactation. Importantly, the lower milk yield of cows that had given birth to male offspring was not compensated for by higher production of protein and fat. The total milk energy production was higher in cows that had given birth to female offspring.

According to the Trivers–Willard hypothesis, natural selection should favour the unequal investment of parents between daughters and sons, depending on the condition of the mother and the reproductive potential of the offspring [58]. The results of the present study can therefore be considered to support that hypothesis. However, according to Fujita et al. [59], in human polygynous populations (equids are also polygynous: authors’ note), in which males have greater reproductive variance than females, mothers in good condition are expected to invest more in sons, while mothers in poor conditions will invest more in daughters. Similarly, Landete-Castillejos et al. [60], in an analysis of the quantity and quality of the milk produced by free-living female Iberian red deer (*Cervus elaphus hispanicus*), found that mothers of males produced more milk, including more protein, fat, and lactose, than mothers of females. The mares included in the present study were in good condition, which suggests that they should offer milk with a better composition to colts, and not, as it turned out, to fillies. However, it seems that the discrepancy between our results and those cited may be due to the fact that the mares were kept in unnatural herd conditions, e.g., without the presence of a stallion [61].

## 5. Conclusions

This study showed that each of the factors analysed (access/lack of access to pasture, lactation number, and sex of the foal) influenced the nutritional value of milk from Sokólski mares, including the content of basic milk components, whey protein profile, and fatty acid profile. The results provide useful information for those managing herds, i.e., that while the use of pasture by horses is an added value in terms of their welfare (which is commonly known), it does not significantly improve the nutritional value of milk, including the content of biologically active substances. It is also worth bearing in mind that the age structure of mares may influence the nutritional value of the milk they produce, which in turn may affect the growth and development of the offspring. In addition, there is some indication as to which sex of offspring is favoured by mares as domesticated animals kept in good condition but in unnatural herd conditions, in order to achieve reproductive success. However, this is preliminary research that should be continued in a larger group of mares.

## Figures and Tables

**Table 1 animals-13-01152-t001:** Diets for mares.

Diet 1	Diet 2
hay—25 kg oats—3–4 kg straw—about 5 kg mineral and vitamin supplements during milking about 0.5 kg crushed oats + carrots	pasture forage—8–9 h ad libitum hay—about 15 kg oats—3–4 kg straw—about 3 kg mineral and vitamin supplements during milking about 0.5 kg crushed oats + carrots

**Table 2 animals-13-01152-t002:** Chemical composition of horse milk (%).

Variable		Pasture		Lactation Number			Foal Gender	
		No	Yes	1	2–3	4–6	Filly	Colt
N Milk Samples	SE	48	31	24	24	31	45	34
		$\bar{x}$ **± SD**	$\bar{x}$ **± SD**	$\bar{x}$ **± SD**	$\bar{x}$ **± SD**	$\bar{x}$ **± SD**	$\bar{x}$ **± SD**	$\bar{x}$ **± SD**
Crude protein	0.025	1.83 ^B^ ± 0.38	1.99 ^A^ ± 0.27	1.82 ^b^ ± 0.29	1.86 ^ab^ ± 0.39	2.00 ^a^ ± 0.35	2.02 ^a^ ± 0.36	1.81 ^b^ ± 0.37
Fat	0.029	0.33 ^A^ ± 0.15	0.21 ^B^ ± 0.13	0.31 ^ab^ ± 0.23	0.21 ^b^ ± 0.25	0.41 ^a^ ± 0.31	0.39 ^A^ ± 0.23	0.26 ^B^ ± 0.13
Lactose	0.027	6.76 ^a^ ± 0.24	6.63 ^b^ ± 0.25	6.77 ^AB^ ± 0.42	6.83 ^B^ ± 0.42	6.53 ^A^ ± 0.61	6.57 ^B^ ± 0.53	6.80 ^A^ ± 0.46
Dry matter	0.040	9.25 ^a^ ± 0.67	9.15 ^b^ ± 0.52	9.24 ± 0.60	9.21 ± 0.47	9.29 ± 0.61	9.29 ^A^ ± 0.52	9.18 ^B^ ± 0.47
Ash	0.003	0.34 ^a^ ± 0.08	0.33 ^b^ ± 0.05	0.34 ± 0.06	0.33 ± 0.08	0.33 ± 0.06	0.34 ± 0.06	0.34 ± 0.07

^a^, ^b^—differences between groups within a factor are significant at *p* ≤ 0.05; ^A^, ^B^—significant at *p* ≤ 0.01.

**Table 3 animals-13-01152-t003:** Whey protein profile in horse milk (% of total proteins).

Whey Protein		Pasture		Lactation Number			Foal Gender	
		No	Yes	1	2–3	4–6	Filly	Colt
N Milk Samples	SE	48	31	24	24	31	45	34
		$\bar{x}$ **± SD**	$\bar{x}$ **± SD**	$\bar{x}$ **± SD**	$\bar{x}$ **± SD**	$\bar{x}$ **± SD**	$\bar{x}$ **± SD**	$\bar{x}$ **± SD**
Percentage of whey proteins in total protein	0.316	54.97 ^A^ ± 5.88	52.01 ^B^ ± 3.81	55.08 ± 5.14	52.90 ± 5.97	52.97 ± 4.53	53.64 ± 4.12	53.69 ± 6.04
including:								
β-Lactoglobulin (β-Lg)	0.195	20.92 ^A^ ± 3.73	18.55 ^B^ ± 2.09	20.54 ± 3.37	19.89 ± 3.49	19.26 ± 3.06	19.42 ± 2.88	20.25 ± 3.61
α-Lactalbumin (α-La)	0.127	15.49 ^A^ ± 2.49	14.78 ^B^ ± 1.85	15.79 ^a^ ± 2.25	14.76 ^b^ ± 1.92	14.95 ^ab^ ± 2.42	15.29 ± 2.27	15.08 ± 2.25
Total immunoglobulins (Ig)	0.089	2.28 ^A^ ± 1.50	1.49 ^B^ ± 0.52	1.82 ± 0.70	2.07 ± 0.82	1.93 ± 1.81	1.87 ± 1.65	1.99 ± 0.79
Lactoferrin (Lf)	0.040	2.20 ^A^ ± 0.66	1.86 ^B^ ± 0.63	2.02 ^AB^ ± 0.55	2.26 ^A^ ± 0.66	1.91 ^B^ ± 0.74	1.79 ^B^ ± 0.57	2.25 ^A^ ± 0.67
Serum albumin (SA)	0.048	1.74 ^B^ ± 0.89	2.18 ^A^ ± 0.70	1.98 ± 0.61	2.07 ± 1.02	1.78 ± 0.86	1.78 ^B^ ± 0.76	2.05 ^A^ ± 0.88
Lysozyme (Lz)	0.106	12.34 ^B^ ± 1.53	13.14 ^A^ ± 2.15	12.9 ^A^ ± 1.83	11.84 ^B^ ± 1.66	13.17 ^A^ ± 1.84	13.52 ^A^ ± 1.93	12.05 ^B^ ± 1.54

^a^, ^b^—differences between groups within a factor are significant at *p* ≤ 0.05; ^A^, ^B^—significant at *p* ≤ 0.01.

**Table 4 animals-13-01152-t004:** Fatty acid profile of horse milk (g/100 g total fatty acids).

Fatty acid		Pasture		Lactation Number			Foal Gender	
		No	Yes	1	2–3	4–6	Filly	Colt
N Milk Samples	SE	48	31	24	24	31	45	34
		$\bar{x}$ **± SD**	$\bar{x}$ **± SD**	$\bar{x}$ **± SD**	$\bar{x}$ **± SD**	$\bar{x}$ **± SD**	$\bar{x}$ **± SD**	$\bar{x}$ **± SD**
C4:0	0.020	0.025 ^B^ ± 0.019	0.222 ^A^ ± 0.324	0.062 ± 0.080	0.072 ± 0.158	0.192 ± 0.093	0.067 ^B^ ± 0.128	0.187 ^A^ ± 0.326
C6:0	0.006	0.108 ^B^ ± 0.036	0.184 ^A^ ± 0.090	0.135 ± 0.049	0.126 ± 0.076	0.151 ± 0.087	0.135 ± 0.063	0.141 ± 0.081
C8:0	0.042	1.460 ^B^ ± 0.593	1.858 ^A^ ± 0.771	2.039 ^A^ ± 0.631	1.386 ^B^ ± 0.739	1.459 ^B^ ± 0.548	1.819 ^A^ ± 0.696	1.470 ^B^ ± 0.660
C10:0	0.129	4.918 ± 1.836	5.178 ± 2.329	6.453 ^A^ ± 1.671	4.770 ^B^ ± 1.934	4.042 ^B^ ± 1.749	5.181 ± 2.159	4.905 ± 1.958
C10:1	0.039	0.934 ^b^ ± 0.451	1.124 ^a^ ± 0.702	1.263 ^A^ ± 0.573	1.061 ^A^ ± 0.399	0.759 ^B^ ± 0.590	1.097 ± 0.651	0.945 ± 0.497
C12:0	0.170	6.837 ± 2.845	6.672 ± 2.380	8.193 ^A^ ± 2.277	7.196 ^A^ ± 2.406	5.249 ^B^ ± 2.398	6.375 ± 2.745	7.063 ± 2.577
C14:0	0.149	7.843 ± 2.661	7.790 ± 1.661	8.324 ^A^ ± 1.933	8.701 ^A^ ± 2.279	6.693 ^B^ ± 2.190	7.051 ^B^ ± 2.132	8.391 ^A^ ± 2.280
C14:1	0.018	0.659 ^A^ ± 0.366	0.433 ^B^ ± 0.233	0.472 ^B^ ± 0.207	0.785 ^A^ ± 0.429	0.474 ^B^ ± 0.253	0.386 ^B^ ± 0.156	0.704 ^A^ ± 0.371
C15:0	0.006	0.265 ± 0.050	0.264 ± 0.166	0.237 ^b^ ± 0.093	0.263 ^ab^ ± 0.100	0.289 ^a^ ± 0.128	0.231 ^B^ ± 0.119	0.289 ^A^ ± 0.099
C16:0	0.309	25.609 ± 3.598	26.367 ± 5.492	24.976 ^B^ ± 3.398	27.268 ^A^ ± 4.745	25.586 ^AB^ ± 4.768	24.392 ^B^ ± 4.336	27.033 ^A^ ± 4.214
C16:1n-9	0.018	0.835 ^A^ ± 0.221	0.290 ^B^ ± 0.127	0.608 ± 0.362	0.680 ± 0.284	0.576 ± 0.328	0.519 ^B^ ± 0.345	0.691 ^A^ ± 0.295
C16:1n-7	0.018	7.619 ^A^ ± 1.819	5.741 ^B^ ± 1.457	6.424 ^B^ ± 1.147	7.581 ^A^ ± 2.446	6.663 ^B^ ± 1.801	6.430 ^B^ ± 1.773	7.195 ^A^ ± 1.961
C17:0	0.005	0.132 ^B^ ± 0.032	0.261 ^A^ ± 0.109	0.157 ^b^ ± 0.058	0.182 ^ab^ ± 0.132	0.201 ^a^ ± 0.084	0.194 ± 0.092	0.176 ± 0.099
C17:1	0.008	0.354 ^B^ ± 0.083	0.533 ^A^ ± 0.130	0.405 ^B^ ± 0.126	0.376 ^B^ ± 0.103	0.483 ^A^ ± 0.148	0.465 ^A^ ± 0.104	0.397 ^B^ ± 0.150
C18:0	0.160	1.320 ^B^ ± 0.400	4.766 ^A^ ± 2.902	2.206 ± 1.092	2.419 ± 2.870	3.321 ± 2.910	2.933 ± 2.531	2.518 ± 2.489
C18:1n-9	0.332	20.785 ^A^ ± 5.528	17.903 ^B^ ± 4.023	18.662 ^B^ ± 4.281	18.271 ^B^ ± 4.694	21.548 ^A^ ± 5.677	20.652 ^a^ ± 5.764	18.885 ^b^ ± 4.566
C18:1n-7	0.042	1.665 ± 0.443	1.649 ± 0.983	1.522 ± 0.463	1.712 ± 0.733	1.729 ± 0.835	1.771 ± 0.745	1.576 ± 0.669
C18:2n-6 (LA)	0.291	11.156 ^A^ ± 3.489	8.724 ^B^ ± 2.462	9.823 ± 3.005	9.852 ± 3.184	10.758 ± 3.668	10.557 ± 3.499	9.912 ± 3.197
C18:3n-6 (GLA)	0.03	0.020 ^B^ ± 0.011	0.067 ^A^ ± 0.032	0.039 ± 0.025	0.031 ± 0.031	0.045 ± 0.037	0.047 ^A^ ± 0.032	0.033 ^B^ ± 0.031
C18:3n-3 (ALA)	0.193	7.120 ^B^ ± 2.504	9.257 ^A^ ± 3.028	7.472 ^B^ ± 2.000	6.895 ^B^ ± 2.854	9.258 ^A^ ± 3.143	9.093 ^A^ ± 2.768	7.144 ^B^ ± 2.746
C18:2 *cis*9 *trans*11 (CLA)	0.008	0.087 ^B^ ± 0.033	0.224 ^A^ ± 0.130	0.156 ± 0.115	0.115 ± 0.059	0.150 ± 0.131	0.184 ^A^ ± 0.134	0.110 ^B^ ± 0.073
C20:0	0.009	0.032 ^B^ ± 0.011	0.246 ^A^ ± 0.156	0.105 ^AB^ ± 0.093	0.069 ^B^ ± 0.075	0.166 ^A^ ± 0.198	0.130 ± 0.099	0.108 ± 0.170
C20:1	0.006	0.206 ^b^ ± 0.063	0.246 ^a^ ± 0.154	0.254 ^a^ ± 0.102	0.196 ^b^ ± 0.107	0.217 ^ab^ ± 0.114	0.279 ^A^ ± 0.097	0.180 ^B^ ± 0.100
∑ SFA	0.639	48.538 ^B^ ± 9.272	53.793 ^A^ ± 7.981	52.863 ^B^ ± 6.851	52.427 ^B^ ± 9.228	47.333 ^A^ ± 9.834	48.500 ^B^ ± 9.669	52.208 ^A^ ± 8.415
∑ MUFA	0.350	33.057 ^A^ ± 5.667	27.920 ^B^ ± 4.607	29.608 ^b^ ± 4.354	30.662 ^ab^ ± 5.621	32.449 ^a^ ± 6.753	31.600 ± 6.926	30.572 ± 4.859
∑ PUFA	0.389	18.382 ± 4.674	18.272 ± 4.418	17.490 ^B^ ± 3.474	16.893 ^B^ ± 4.750	20.212 ^A^ ± 4.623	19.88 ^A^ ± 4.123	17.199 ^B^ ± 4.553
n-6 PUFA	0.291	11.175 ^A^ ± 3.494	8.791 ^B^ ± 2.474	9.861 ± 3.004	9.883 ± 3.188	10.804 ± 3.660	10.604 ± 3.488	9.945 ± 3.201
n-3 PUFA	0.193	7.120 ^B^ ± 2.504	9.257 ^A^ ± 3.028	7.472 ^B^ ± 2.000	6.895 ^B^ ± 2.854	9.258 ^A^ ± 3.143	9.093 ^A^ ± 2.768	7.144 ^B^ ± 2.746

LA—linoleic acid; GLA—γ-linolenic acid; ALA—α-linolenic acid; CLA—conjucted linolic acid; SFA—saturated fatty acids; MUFA—monounsaturated fatty acids; PUFA—polyunsaturated fatty acids; ^a^, ^b^—differences between groups within a factor are significant at *p* ≤ 0.05; ^A^, ^B^—significant at *p* ≤ 0.01.

**Table 5 animals-13-01152-t005:** Parameters characterizing the fatty acid profile of milk (g/100 g total fatty acids).

Parameter		Pasture		Lactation Number			Foal Gender	
		No	Yes	1	2–3	4–6	Filly	Colt
N Milk Samples	SM	48	31	24	24	31	45	34
		$\bar{x}$ **± SD**	$\bar{x}$ **± SD**	$\bar{x}$ **± SD**	$\bar{x}$ **± SD**	$\bar{x}$ **± SD**	$\bar{x}$ **± SD**	$\bar{x}$ **± SD**
n-6/n-3 ratio	0.039	1.684 ^A^ ± 0.630	1.032 ^B^ ± 0.391	1.421 ± 0.592	1.573 ± 0.600	1.305 ± 0.675	1.286 ^b^ ± 0.648	1.526 ^a^ ± 0.606
DFA (desirable fatty acids)	0.597	52.760 ± 9.228	50.958 ± 6.593	49.304 ^B^ ± 6.746	49.973 ^B^ ± 8.272	55.982 ^A^ ± 8.088	54.414 ^A^ ± 8.296	50.289 ^B^ ± 7.904
HSFA (hypercholesterolaemic saturated fatty acids)	0.517	40.289 ± 8.026	40.828 ± 6.141	41.493 ^A^ ± 6.152	43.166 ^A^ ± 7.363	37.528 ^B^ ± 7.219	37.819 ^B^ ± 6.788	42.487 ^A^ ± 7.089
AI (atherogenicity index)	0.034	1.340 ± 0.571	1.478 ± 0.537	1.474 ^A^ ± 0.453	1.563 ^A^ ± 0.584	1.193 ^B^ ± 0.567	1.244 ^B^ ± 0.550	1.507 ^A^ ± 0.543
TI (thrombogenicity index)	0.022	0.850 ± 0.308	0.908 ± 0.450	0.858 ^AB^ ± 0.231	1.008 ^A^ ± 0.430	0.775 ^B^ ± 0.385	0.752 ^B^ ± 0.366	0.962 ^A^ ± 0.350

^a^, ^b^—differences between groups within a factor are significant at *p* ≤ 0.05; ^A^, ^B^—significant at *p* ≤ 0.05.

## Data Availability

The data presented in this study are available on request from the corresponding author.

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
