# Peer review of "The Influence of Selected Factors on the Nutritional Value of the Milk of Cold-Blooded Mares: The Example of the Sokólski Breed"

_animals, 2023, doi:10.3390/ani13071152_

Round 1

Reviewer 1 Report

line 64 - sport is not a field of us of draft horses 

line 66 - maybe not the most important food - but everyday food or part of everyday diet

line 67 - diarrhoea is not very common after fresh mares milk consumption, fermentation is the processing of milk to prolong the usage of the milk

line 68 - it is not just alcoholic fermentation 

line 117 4*4 m boxes. (not stalls?)

Statistical model lines 221 to 226

for indices have to use lower case function, for fixed effect w, you use index I, so i=2 (not n=2), the same is for other fixed effects (j=2 and k=3), there is also confusion in the statistical model as why you use 'l' as a symbol of fixed effect of lactation and as an index. 

A general comment on the statistical model: the effect of the different diets is more than just diet it is also year. It is somehow nested with mare and lactation. The interpretation has to be made with caution. 

results lines 237-244 and also later in the results and in the discussion: please use % beside p.p.

Tables 2-5.

It is not clear which numbers you presented in the tables. According to the statistical model, LSMs and SE were expected. There stay mean and SD. According to the estimation of differences between the levels of each fixed effect Bonferonni test is just suitable for the effect of lactation, for effects with two levels t-test is possible.

General in statistical analysis first in the results Analysis of variance table is expected. It will be very informative how much variability explains each effect at least in the main analysed traits

Author Response

Firstly, we would like to thank the reviewer for the valuable comments and suggestions. We have tried to take them all into account in the revised version of the manuscript. Some of the shortcomings pointed out by the reviewer are obvious and we unfortunately did not notice them. For this we thank you very much.

  • The introduction includes all the reviewer's suggestions;
  • In the chapter "Material and Methods", the word "stalls" has been changed to "boxes"
  • As suggested, the entries in the statistical model and its description have been changed;
  • The year of milk sampling was included in the statistical calculation, unfortunately we did not include in the statistical model, which we have now rectified;
  • In the chapter "Results” the unit % was used instead of p.p.;
  • In the tables, a column has been added with values for standard error of the mean (SE).

Reviewer 2 Report

Dear authors, 

The manuscript is easy to follow and logical. There are some minor suggestions for correction: abbreviations in the abstract such as α-La, β-Lg, SA, Ig, Lf, and Lz need to be fully explained in this section of the manuscript (please, put full terms). The term "the best fatty acid profile" in the abstract needs to be explained. Author Egito et al (2011)  wrote about the susceptibility of equine κ- and β-caseins to hydrolysis by chymosin, not about using kumiss in the treatment. Please reconsider this reference. On page 3, line 123 please change "from" to "between". Analytical methods used must be cited in the list of references (please see attachment). I suggest changing p.p. as a unit into % through the text. On page 5, line 238 please change "without" to "with". Please add "of total protein" at the end of the title of table 3. Please correct the order of numbers of the pages after page 6 (see attachment). When you cite the results of others it is not common to mention statistical significance. If we cite others we take their results as generally true (see attachment line 372). There are some minor mistyping in the text (see attachment). On lines 399-402, reference number 28 must be changed to 29. The same for line 420. Cervus elaphus hispanicus need to be written in italic. 

Sincerely yurs, 

Author Response

First, we would like to thank the reviewer for the comments and suggestions. We have attempted to take all of them into account in the revised version of the manuscript.

  • In the abstract we have given the full names of whey proteins;
  • In the abstract we explained the term “the best fatty acid profile”, completing the sentence with content – “the lowest concentration of SFAs and the highest concentration of MUFAs and PUFAs”;
  • The inclusion of the Reviewer's suggestion in the Abstract and Simple Summary has resulted in exceeding the required word limit (200 words), and we have therefore had to modify it slightly;
  • With regard to the sentence "In Russia, fermented mare milk (koumiss) is used in the treatment of gastrointestinal and cardiovascular disease", the reference from Egito et al. (2001) replaced by Lozovich, S. Medical uses of whole and fermented mare milk in Russia. Cultured Dairy Products Journal (USA) 1995, 30(1), 18-21;
  • all norms covered in the "Material and Methods" chapter are included in the reference list.

All editorial suggestions indicated by the Reviewer have been included in the manuscript.

Reviewer 3 Report

The manuscript describes chemical composition, fatty acid and protein profile of cold-blooded mare's milk and statistical associations with selected management factors.

In the introduction chapter authors offer brief overview of the contemporary horse breeding and use of horses. Authors point out the usage of mare's milk and milk products in human diet as well as the difference between cow's and mare's milk regarding chemical composition and therapeutic properties.

In the material and methods chapter authors describe methods applied to collect and handle the samples. Methods used to analyse samples are described with sufficient details.

A few details in material and methods chapter remain unclear. Hence, authors are asked to additionally explain number of mares included per year since the sum of mares per year (n=15) is unequal with the sum of colts and fillies (n=16) (line 115/116). The discrepancy is probably due to the two mares in same lactations or the twins.

Authors stated that nine mares were enrolled in the research. However total of 16 lactations were followed during the specified time frame. Obviously, several mares were enrolled in the research in more than one lactation. Hence authors are asked to describe how they dealt with repeated or paired data.

Research was carried out on total of 79 milk samples (line 143). The results would be more convincing if the number of samples per year is also stated as well as the distribution of colts and fillies per year since the results might be in fact confounded for example by precipitation in a certain year. If a factor (for example sex of youngsters) is unevenly distributed it can mask the true statistical association with another factor.

The results are presented as text and tables. Tables are not self-explanatory. Hence authors are asked to complete the tables with value of n for each category (Pasture yes=x; no=y...).

In the discussion chapter authors compare their own results with the results of similar analyses carried out elsewhere either in cow's or mare's milk and offer a reasonable explanation for observed discrepancies between.

The conclusions are based on the obtained results but should be shortened without repetition of already mentioned results.

Author Response

First, we would like to thank the reviewer for the valuable comments and suggestions. We have attempted to take all of them into account in the revised version of the manuscript.

  • Regarding the inaccuracies in the chapter "Material and methods" regarding the number of lactations and foals born included in the study, we inform you that the correct value is 15 (7 colts and 8 fillies). In addition, information was provided on how many foals were born in each year of the study (2017 – 2 colts and 2 fillies, 2018 – 2 colts, 2019 – 2 colts and 2 fillies, 2020 – 1 colt and 4 fillies). The chapter has corrected the unfortunate total number of milk samples taken. The correct value is 193. This value has also been corrected in the abstract. The number of milk samples taken in each year of the study has also been given. In the current version, the sentence reads as follows: In total 193 milk samples were collected (in 2017 – 48, 2018 – 40, 2019 – 47 and 2020 – 58 samples).
  • Regarding the reviewer's query on how the authors dealt with repeated or paired data, we report that the authors used the general linear model of variance analysis as the most versatile and robust to data pairing. It was not possible to use a more favourable model such as the analysis of variance model for repeated measures, as successive lactations did not always involve the same mares. In such a setting, an analysis of the fixed effect of the factor of subsequent lactation seemed most favourable.
  • Regarding the comment on the possible influence of additional factors on the results obtained in a given year (e.g. precipitation), we would like to inform you that the mares used the pasture for two years 2019-2020. The composition of the pasture sward, the intensity of use and the general grazing conditions were similar in these years.
  • The tables have been supplemented with n values for each category.
  • The summary has been shortened in order not to repeat the results obtained.